# Reproducibility report on Explainable Automated Fact-Checking for Public Health Claims

## 1    Reproducibility Summary

### 2    Scope of Reproducibility

3    Our work consists of two major parts: (1) Reproducing results from Kotonya & Toni(Authors) (3) (2) Performing
4    experiments to improve test accuracy and other metrics for veracity prediction. We did not use BioBERT(20) model
5    deliberately for veracity prediction as it did not perform well on the defined metrics, as observed in the original paper(3).
6    Authors were doubtful of how good the rouge metric is in conveying the quality of explanations, so they used human
7    evaluation to evaluate the explanations generated. We stuck to rouge score for evaluating the explanations generated.

### 8    Methodology

9    Authors did not publish the code for fine-tuning BERT(10) and SciBERT(7) models for veracity prediction. For
10   explanation generation the authors use a BERT based model which was not made public, so we chose the BART model
11   pre-trained on CNN-DailyMail dataset. We have written a functional and modular code.[1] which is easy to reproduce
12   and comprehend.

### 13    Results

14   The accuracy for veracity prediction using BERT base model (top 5 sentences) was 3% lower than that published by the
15   authors. The accuracy for veracity prediction using SciBERT (top 5 sentences) was 4.73% lower than that published by
16   the authors. SciBERT performed well on all the test metrics for veracity prediction. While the accuracy was close, the
17   macro F1, precision and recall were inconsistent with the authors' claim. For explanation generation, the automated
18   evaluation metric was rouge(21). In case of R1 and RL, we got f1 measure, which was around 30% more than what was
19   mentioned in the paper(3). Improvements were also observed in R2 rouge score. 4.1.4. We also checked some of the
20   explanations that were generated, and the results were up to the mark with gold standard explanations.

### 21    What was easy

22   It was easy to implement the code for veracity prediction using two different BERT models. The model used for
23   summarization was available in the Hugging Face library, pretrained on the same dataset as the authors. Without much
24   effort, we were able to fine-tune the model on our dataset.

### 25    What was difficult

26   The implementation code was not available in author's GitHub repository[2]. We had to implement code ourselves. It
27   was difficult to increase the accuracy of the models to get close to that published by the authors.

### 28    Communication with original authors

29   We tried contacting the authors many times, but unfortunately could not make any contact.

---

[1]https://github.com/saswat01/Reproduce-Health_Fact_Checking
[2]https://github.com/neemakot/Health-Fact-Checking

# 1   Introduction

A great amount of progress has been made in the area of automated fact-checking. This includes more accurate machine learning models for veracity prediction and datasets of both naturally occurring (Wang, 2017; Augenstein et al., 2019; Hanselowski et al., 2019) and human-crafted (Thorne et al., 2018) fact-checking claims, against which the models can be evaluated. We introduce a framework for generating explanations and veracity prediction specific to public health fact-checking. We show that gains can be made through the use of in domain data. The second shortcoming we look to address is the paucity of explainable models for fact-checking (of any kind). Explanations have a particularly important roles to play in the task of automated fact checking, especially in health-related claims where domain specific knowledge is required to understand the context. Explainable models can also aid the end users' understanding as they further elucidate claims and their context[3].

This work intends to perform reproducibility, experiments to improve score on evaluation metrics for veracity prediction and perform an ablation study(remove BioBERT for veracity prediction and human evaluation of generated explanations) and validate the metrics to evaluate the experiments. The work also contributes a pipeline of veracity prediction and explanation generation, using which we can get veracity prediction of a claim and explanation verifying the prediction.

# 2   Scope of reproducibility

As mentioned by the authors, (3) the veracity predictions made on the claims and evidence sentences (top 5) gave the best accuracy when SciBERT was used and BERT(base-uncased) model gave the best precision score. The SciBERT model succeeded on all the test metrics for veracity prediction. But the BERT model did not surpass SciBERT in terms of precision score, which was contrary to what was observed by the authors. We also took a data centric approach and observed the performance of the same transformers on dataset alterations for veracity prediction.

For explanation generation, we used the top 5 sentences returned by SBERT in veracity prediction stage. The BART based summarization model pretrained on CNN/DailyMail dataset, yielded better results than BERT based extractive abstractive summarization model pretrained on the same dataset as mentioned in the paper (3). Our central aim was to reproduce these results as close as possible to the authors. The results of our experimental findings are shown in Table 4.1.1, 4.1.2 and 4.1.3.

# 3   Methodology

We have provided the entire code to fine-tune the BERT, SciBERT and DistillBart transformers for veracity prediction and explanation generation. The processed dataset has been put up in the GitHub repository (1), which can be used straightaway for fine-tuning the transformer models on downstream tasks. The modularity of the code makes it comfortable for experimentation. For, e.g., if an individual wants to increase/decrease hyperparameters like batch size, epochs, learning rate etc. they can easily do that. All the instructions have been specified in the GitHub repository (1) regarding the usage of the repository and experimentation. We have made use of PyTorch Lightning, which makes training swifter. Along with that, we train all the models using early stopping with patience of 2 or 3. While training the models, we save the checkpoints when the validation loss is least. It helps to track and save the model with the best weights.

The fine-tuning was performed on Tesla T4 15.84 Gigabyte GPUs provided on the Google Colaboratory platform. The recommended batch size provided by the authors was 16 only the GPUs could support a batch size of 13. We used batch size 13 for fine-tuning both the transformer models for veracity prediction and a batch size of 8 for fine-tuning DistillBart for explanation generation. Computational time and other details have been provided in Section 3.6. We have also provided a convenient test script which can predict label, select evidences from main text and generate an explanation for the claim text.

## 3.1   Model descriptions

For veracity prediction, we make use of pretrained BERT(base-uncased) model and SciBERT(scibert-scivocab-uncased) model from Hugging Face library. The tokenizers for BERT(base-uncased) and SciBERT(scibert-scivocab-uncased) were also used from the Hugging Face library. For explanation generation, we used DistillBart model and tokenizer pretrained on CNN-DailyMail dataset from the Hugging Face library.

---

[3]Excerpts taken from paper (3)

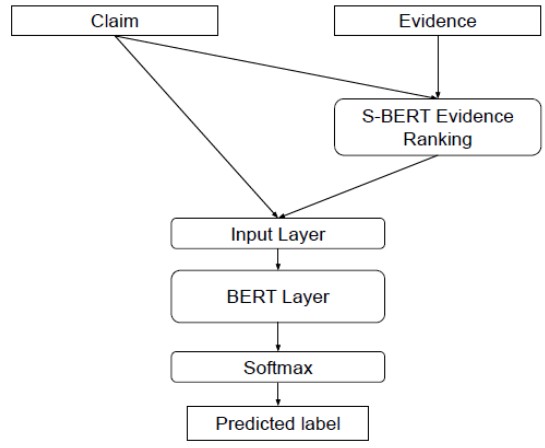

Figure 1: Veracity Prediction model Architecture

1. The pretrained BERT(base-uncased) model has 12-layer, 768-hidden, 12-heads, 110M parameters. It was pretrained on lower-cased English text.

2. The pretrained SciBERT model is a BERT model pretrained on papers taken from Semantic Scholar of Corpus size 1.14M papers and 3.1B tokens.

3. The pretrained DistillBart 12-6 model is pretrained on CNN- DailyMail dataset, which has 300k unique news articles written by journalists at CNN and the Daily Mail. It has 306M parameters.

4. A smaller DistillBart model pretrained on the same CNN-DailyMail dataset. It has 230M parameters.

We have provided the above-mentioned fine-tuned models in the repository (1).

## 3.2 Datasets

The Pubhealth dataset constructed by the authors (3) contains 11,832 claims for fact-checking. The claims were related to several topics like biomedical research, government healthcare policies and other health related stories. The dataset is divided into three splits train, dev and test dataset. Main features in the dataset was claim ID, claim sentence, main text containing article text, explanation, and label. The distribution of labels in train, dev and test dataset is displayed in Figure 2. The dataset had some NA values for some features. We dropped the rows containing NA values. After dropping the rows we had 9806 observations in train dataset, 1235 observations in test dataset and 1217 observations in validation dataset.

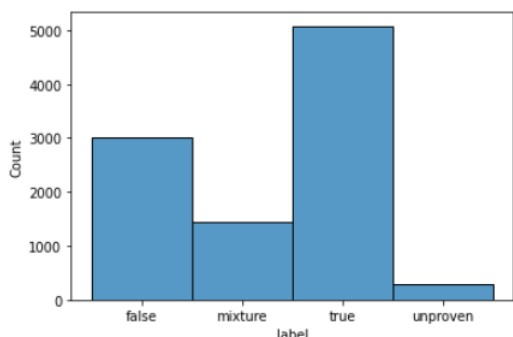

(a) Label distribution for train dataset

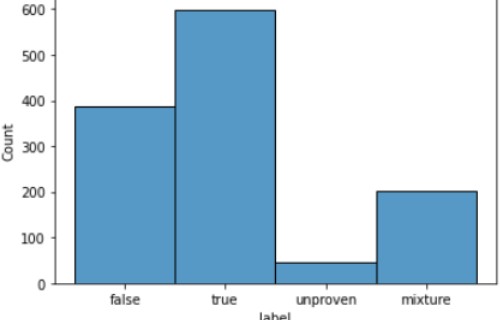

(b) Label distribution for test dataset

Figure 2: Distribution of labels in dataset

We also created a derived dataset using authors code6.3 for preprocessing and then applying SBERT to the main text to get top k sentences for veracity prediction according to their cosine similarity with respect to the contextualized representation of the claim sentence.

The dataset in raw and processed form are available in our GitHub(1) repository.

### 3.3 Hyperparameters

Details regarding the hyperparameters like batch size, number of epochs, learning rate are mentioned in the original research paper by the authors. Although, it helped us to reproduce the paper, information about other hyperparameters (for e.g., token length) and detailed methodology for fine-tuning task for veracity prediction was not offered. We used the same learning rate as mentioned by the authors, i.e., 1e-6. As was mentioned earlier, the GPU provided on the Google Colaboratory platform could not support a batch size of 16 (recommended by the authors) we used batch sizes of 8, 10, 12 and 13 for experimentation. Out of which, a batch size of 13 established the best results; a batch size of 12 also showed synonymous results. We altered epochs from 4 to 7 and got 5 as the most appropriate number of epochs. Hence, the number of epochs for fine-tuning the models was altered from 4 (recommended by the authors) to 5 as it gave best results in our case. Encode plus tokenizer with maximum length of 512 was used for fine-tuning both the transformers. We optimized our model using Cross-Entropy Loss (recommended by authors) for veracity prediction.

For all experiments related to veracity prediction, hyperparameter trials were done for each possibility of batch sizes {10, 12, 13} and number of epochs {4, 5, 7}.

In the case of explanation generation, authors didn't give any details for hyperparameter tuning. We enforced manual search with a couple of combination of hyperparameter values. We used 5e-5 as the learning rate, batch size of 8, number of epochs was 3. We used Adam W optimizer with maximum input length of 512 and the maximum output length was 128.

### 3.4 Experimental setup and code

For veracity prediction models were evaluated on test dataset using macro F1, precision, recall, and accuracy metrics. Linear scheduler with warm up was used to train the language models to decrease chances of early over-fitting or skewness. For cleaning the top 5 evidence sentences, we used regex library provided by the Python language. Precise instructions have been provided in the repository(1) to assist you to train the models and test them, along with discerning the test metrics.

For explanation generation, the performance on the test dataset was assessed using rouge score metrics. The training script is easy to run aiding arguments, with option to save the model which can be used later to evaluate on the test data. We have provided a test script in the GitHub repository, which lets you perform veracity prediction and generate explanation for the claim sentence simultaneously.

### 3.5 Extended Experiments

Apart from reproducing the paper, experiments were conducted, to answer the following questions:

- Can managing class imbalance in the dataset lead to better performance of models on evaluation metrics for veracity prediction ?
- Can text cleaning improve model performance for veracity prediction ?
- Can any other hyperparameter improve model performance for veracity prediction ?

To handle class imbalance in the dataset, synonym matching (replacing maximum 15 words in the top 5 evidence sentences using synonyms) was used as the augmentation technique. Most observations lied under the "True" label, so we made the "Mixture" and "Unproven" label observations equivalent to the "False" label, which had observations next to the "True" label. We conducted experiments on the augmented dataset using the BERT model, 4.1.2 discusses the results. The SciBERT model did not give satisfactory results on the augmented data, so we did not perform extensive experimentation utilizing it.

Text cleaning was performed using the regex library provided by the Python language. We removed redundant text, i.e., square brackets, links, punctuation, and words containing numbers from the top 5 evidence sentences. The SciBERT model fine-tuned on the clean top 5 evidence sentences performed flawlessly on all the evaluation metrics 4.1.3.

It was observed that the average token length was 125 and a maximum token length of 512 was observed. There were a handful of tokens whose token length was 512. We reduced the token length from 512 to 350 to fine-tune BERT and

SciBERT on the clean data. We chose 350 as the trusted token length, as it was capable of representing the token length of 95% of the sentences. It gave the best results and the results are discussed in Section 4. A 2% gain in accuracy was discovered when the SciBERT model was fine-tuned on clean top 5 evidence sentences along with a token length of 350.

## 3.6 Computational requirements

As the experiments are done on an easily accessible environment, there are no specific requirements one needs to reproduce and implement our work. You need a laptop/PC and an internet connection to perform everything that we have published in the report and in the GitHub repository (1).

The computational time for various models and other relevant details have been provided in the tables below.

| Model | Train time(in sec.) | Train time(in hours) |
|---|---|---|
| BERT(epoch 4) | 3420 sec | 0.95 hours |
| BERT(epoch 5) | 5400 sec | 1.5 hours |
| BERT(epoch 7) | 7200 sec | 2 hours |
| SciBERT(epoch 4) | 3600 sec | 1 hours |
| SciBERT(epoch 5) | 5940 sec | 1.65 hours |

Table 1. Training computational time for veracity prediction

| Model | Test time(in sec.) | Test time(in minutes) |
|---|---|---|
| distilbart-cnn-12-6 (epoch 3) | 6300 sec | 1.75 hours |
| distilbart-cnn-6-6 (epoch 3) | 4500 sec | 1.25 hours |

Table 2. Training computational time for summarization

# 4 Results

For calculation of all the metrics for veracity prediction, Scikit-learn library was used. The SciBERT model gives the best accuracy, F1 score, precision, and recall on the Pubhealth dataset for veracity prediction. It supports the original claim of the authors except that their BERT model gave better precision than SciBERT which was not observed in our experiments. Also, from all the experiments we conducted, SciBERT model gave the best results when the top 5 evidence sentences was cleaned and the token length was shrunk. BERT model gave the best results when it was trained using the best hyperparameters, as discussed 3.3. Also, it was observed that BERT results were not very different when it was fine-tuned on clean top 5 evidence sentences. Distillbart based model gave better results than ExplanerFC-Expert

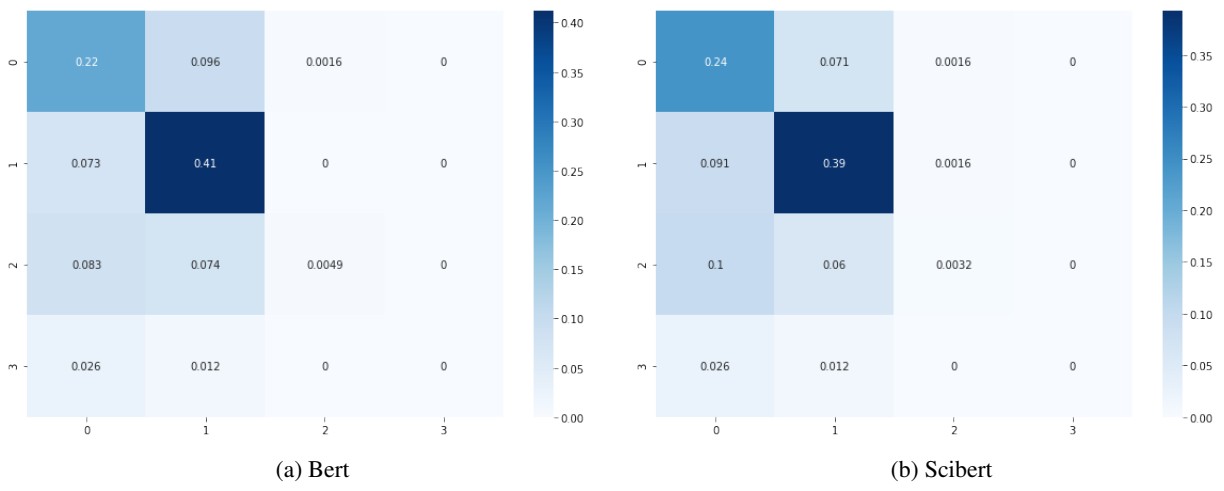

(a) Bert       (b) Scibert

Figure 3: (a)BERT and (b)SciBERT confusion matrix

model used by authors. The rogue scores are given in the table 4.1.4 along with authors' best model.

### 4.1 Results reproducing original paper

It can be comprehended that SciBERT surpasses BERT on every evaluation metric in section 4.1.1.

#### 4.1.1 Veracity prediction best results

Both models were fine-tuned using the best hyperparameters, except the token length. It can be sighted that accuracy metrics is proximate to the original assertions of the authors but the precision, accuracy and F1 score is contrary from what was originally verified by the authors. SciBERT performs unexcelled on all the metrics for veracity prediction. It partially supports the authors' assertions, as the precision of the BERT model is not better than the SciBERT model.

| Model | Pr. | Rc. | F1 | Acc. |
|---|---|---|---|---|
| BERT(top 5) | 0.35 | 0.39 | 0.35 | 63% |
| **SciBERT(top 5)** | **0.44** | **0.41** | **0.37** | **65%** |

Table 3. BERT fine-tuned using token length 512, SciBERT fine-tuned using token length 350

#### 4.1.2 Augmentation Result using BERT(top 5) with batch size 12

We experiment the phenomenon of epoch size on fine-tuning the BERT model on the dataset for veracity prediction. Text augmentation improved the precision, recall and F1 score of the BERT model but was not improving the accuracy of the model. Data augmentation also facilitated the BERT model to categorize the labels more accurately, particularly the "Mixture" and "Unproven" labels.

| Epochs | Pr. | Rc. | F1 | Acc. |
|---|---|---|---|---|
| 4 | 0.41 | 0.39 | 0.37 | 57% |
| 5 | 0.42 | 0.35 | 0.34 | 59% |
| 7 | 0.45 | 0.37 | 0.35 | 59% |

Table 4. BERT metrics when trained on augmented dataset using synonym replacement, token length 512

#### 4.1.3 Text Cleaning Result on language models for veracity prediction

| Model | Pr. | Rc. | F1 | Acc. |
|---|---|---|---|---|
| BERT | 0.31 | 0.38 | 0.34 | 62% |
| SciBERT | 0.44 | 0.41 | 0.37 | 65% |

Table 5. Language models trained using the best hyperparameters with token length 350

#### 4.1.4 Results for explanation generation with batch size 8

The metric used to evaluate the explanation quality was rouge.

| Model | Metric | precision | recall | F1 |
|---|---|---|---|---|
| distilbart-cnn-12-6 | R1 | 0.472 | 0.451 | 0.461 |
|  | R2 | 0.181 | 0.173 | 0.177 |
|  | RL | 0.409 | 0.392 | 0.4 |
| distilbart-cnn-6-6 | R1 | 0.459 | 0.438 | 0.447 |
|  | R2 | 0.165 | 0.158 | 0.161 |
|  | RL | 0.395 | 0.377 | 0.385 |
| ExplanerFC-Expert | R1 | - | - | 0.323 |
|  | R2 | - | - | 0.135 |
|  | RL | - | - | 0.27 |

Table 6. Explanation generation results

## 5 Ablation Study

As discussed in the original research paper (3), BioBERT v1.1 and BioBERT v1.0 did not show any significant performance on the metrics mentioned for veracity prediction. We excluded the BioBERT transformer model for the reproducibility of veracity prediction.

For evaluation of explanations, authors used two methods. The second one was human evaluation, in which authors asked humans to assess the quality of the gold and generated explanations. We could not do human evaluation of explanations. Authors were skeptical about how good rouge score comprehends the usefulness or the quality of the explanations, so they also performed human evaluations of generated explanations. Authors calculated coherence to assess the quality of the explanations. We stuck to rouge metric for our exclusive evaluation criteria for explanation generation.

# 6 Discussion

The experimental results discussed above supports the overall claims by authors. For veracity prediction, we could reproduce the results claimed by authors for SciBERT model by using clean top 5 evidence sentences and token length of 350. These details were not provided in the paper and were revealed using different experimental setups. Unfortunately, we could not connect to the authors to substantiate our methodology, but the experimental results convinced us to conclude this approach suitable. We could have also experimented with different types of augmentation techniques to discover how the models would have performed. Also, we could have made the maximum token length closer to the average token length to record the empirical observations of model performance. It may have been possible that experimenting on these varied scenarios would have concurred in metrics closer to that published by the authors.

For the explanation generation task, the authors used two different types of evaluation method. One was automatic evaluation using rouge metric. Rouge metric is considered to be the best metric when it comes to summarization tasks. As discussed in the ablation study part, authors did use human evaluation. We could have increased the k value above 5 to check if that generates better results. We used 128 as the max output token, as the average length of gold standard explanations were more or less the same.

## 6.1 What was easy

As the authors provided the script in their GitHub repository6.3 to extract top 5 evidence sentences from main text, it helped us a lot to kick-start the implementation of transformer models for veracity prediction. Also, the authors provided a clear architecture[1] for veracity prediction that helped us to understand the flow of the whole process for veracity prediction.

## 6.2 What was difficult

The fine-tuning procedure was not explained in detail by the authors, due to which it took significant amount of time and experiments to search the most suitable hyperparameters for fine-tuning the models for veracity prediction. The instructions provided by the authors about the abstractive-extractive model which they used were difficult to follow.

## 6.3 Communication with original authors

We tried reaching out to the authors by email. Unfortunately, we could not connect to them. We apprehend they maybe busy. We have also sent them this report for verification and expect their response.

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
