# OpenReview forum: "Reproducibility report on Explainable Automated Fact-Checking for Public Health Claims"
_ML_Reproducibility_Challenge/2021/Fall — Reject_

### Official Review · Reviewer_Q5Vz · 2022-02-22
**Straingforward reproduction study, some useful results/information, but reporting needs substantial improvements.**

**Rating:** 6
**Confidence:** 4

**Review:**

This paper reports on reproduction of "Explainable Automated Fact-Checking for Public Health Claims" (EMNLP, Kotonya and Toni, 2021). A subset of the results were reproduced (mainly, the reproduction study omits the human evaluation). The reproduction results in some substantial (?) differences between the original study and the reproduction effort.

Part of the experiments described in the paper (fine-tuning some pre-trained language models) was (re)implemented during the reproduction study.

One of the main issues with the present report is the clarity.  Although the language is understandable with some effort, there are numerous language and typography mistakes that make some of the points difficult to understand for the reader. I only list some of them below. If I listed all that I see, the list would be much longer.

Besides this a few notes on reporting of the reproduction effort:

- In my opinion, some decisions are not motivated well. For example, omitting the experiments with BioBERT because the original study found it to perform worse than the others. In fact, this could motivate the reproduction study to verify such conclusions.
- It would help the reader understand the reproduction success/failure if the original scores and the ones reproduced were presented side by side.
- Although it is difficult to read otherwise (see below for more), the results in Figure 3 indicate a clear majority bias. Additional experiments to overcome this problem may have been interesting.


The following is an incomplete list of minor issues, mainly about
presentation.

- The reference for the original paper reproduced is wrong (the reference points to a survey article by the same authors).
- Better leave space between a opening parenthesis/bracket and the preceding word.
- The paper uses, an unusual citation format (numeric references with round brackets).
- line 7: "stuck to" -> "stuck with" (but probably also a bit too informal for a formal report)
- line 11: remove period from "modular code.1 which is"
- line 17/18: Better use consistent notation for "F1"
- Footnote marks should be placed after punctuation.
- Providing a non-anonymized repository breaks the anonymity of the submission.
- line 37: roles -> role
- footnote 3: you need to be more explicit about the sections quoted from the original paper.
- line 45: " the authors, (3) the ..."  -> " the authors (3), the ..."
- line 51: remove comma in "on CNN/DailyMail dataset, yielded "
- line 54: the references provided are not tables
- line 59: "For, e.g.," -> "For example,", or "E.g.,"
- Section 3.1: you should consider citing all the models mentioned.
- line 76-82: the information provided for each model should be consistent (e.g., number o layers, training data).
- line 79: "1.14M papers" documents?
- line 87: "claim ID" is not a feature, it maybe part of the file format, but you should be worried if it is an informative feature
- Figure 2: best to keep the order of classes the same on side-by-side figures.
- line 92: The reader likely cannot not understand what "authors code6.3"  means.
- line 103: "synonymous" -> "same" or (more likely) "similar"
- The style (particularly in section 3.4) is too conversational.  Referring to the reader directly is unusual in academic papers.  - line 132: 'next to the "True" label' -> "similar number of True labels"?
- line 137: expressions like "flawlessly" is a bit too informal, and subjective unless quantified.
- line 138: "observed" repeated
- line 154: you should cite the libraries used (here scikit-learn)
- line 159: Distilbart -> DistilBART,  ExplanerFC ->  ExplainerFC
- figure 3: the confusion matrices are confusing for this reader: (1) how to map the numeric class labels to actual ones? (2) what are the numbers presented in the cells, they are not whole numbers, but they do not seem to be normalized on any of the axes. I think absolute numbers would be much easier to understand.
- Table 3: Not impossible, but the macro-averaged substantially F1 is lower than both precision and recall for SciBERT (more likely to be between them). It is worth checking if this is a reporting error.
- line 173: you should clarify why do you want to improve accuracy.  Improving accuracy on this imbalanced data set is probably what we want.
- Section 5: "an ablation study" is where you remove parts of the model/system and do additional experiments. When you decide not to do some experiments, it wouldn't be called an ablation study.
- The bibliography contains many items that are not referenced in the main text.
- I recommend referring to the published versions of the papers (e.g., EMNLP for the original paper), rather than preprint archives.

---

### Official Review · Reviewer_sp1U · 2022-03-01
**Reproducibility Study on "Explainable Automated Fact-Checking for Public Health Claims"**

**Rating:** 5
**Confidence:** 2

**Review:**

This submission is based on reproducing the work "Explainable Automated Fact-Checking for Public Health Claims". The authors had difficulty in reproducing the results of the paper. They also could not get in touch with the authors of the original work being replicated, but have shared the document with them for any feedback.

In this work, the authors have done a great effort in trying to implement the work from scratch as there wasn't a openly published source code and some of the details of the training procedure and models were not present in the original paper. The authors have made their code modular and easier to understand, but are not sure if their methodology exactly matches the original work's methodology due to a missing original reference to the work being replicated.

A few things that the authors might do to improve their submission:
1. Restructuring of some of the parts of the document might help in providing the reader a better high-level understanding of the original paper. Some background on the task at hand, the choices made in the original paper and an overview of the approach might help provide a better flow in understanding the problem and the challenges. This can be easily added even without going into practical details of the paper.
2. Some more ablation studies could be added. Although since a repo of the original work wasnt available, and the original authors could not be contacted, this could be challenging, but some of these ablations might actually help answer and fill in some of the gaps.
3. There are some typos across the document, and this should be an easy fix to improve the overall writing.

The authors of this submission seem to have done great work, and their efforts are encouraging. Filling in the gaps once the original authors could be contacted, adding more ablation studies, and restructuring parts of the document to improve the writing should significantly help in improving the overall future score of the submission.

---

### Meta-Review · Program_Chairs · 2022-04-09

**Recommendation:** Reject
**Confidence:** 5

**Metareview:**

The reviewers have identified a number of areas of improvement within the submission.  Consequently, the submission is not accepted.

---

### Decision · Program_Chairs · 2022-04-09

Reject